# Flavin-dependent halogenases catalyze enantioselective olefin halocyclization

Dibyendu Mondal [1], Brian F. Fisher [1,2], Yuhua Jiang[1] & Jared C. Lewis [1✉]

Halocyclization of alkenes is a powerful bond-forming tool in synthetic organic chemistry and a key step in natural product biosynthesis, but catalyzing halocyclization with high enantioselectivity remains a challenging task. Identifying suitable enzymes that catalyze enantioselective halocyclization of simple olefins would therefore have significant synthetic value. Flavin-dependent halogenases (FDHs) catalyze halogenation of arene and enol(ate) substrates. Herein, we reveal that FDHs engineered to catalyze site-selective aromatic halogenation also catalyze non-native bromolactonization of olefins with high enantioselectivity and near-native catalytic proficiency. Highly selective halocyclization is achieved by characterizing and mitigating the release of HOBr from the FDH active site using a combination of reaction optimization and protein engineering. The structural origins of improvements imparted by mutations responsible for the emergence of halocyclase activity are discussed. This expansion of FDH catalytic activity presages the development of a wide range of biocatalytic halogenation reactions.

[1] Department of Chemistry, Indiana University, Bloomington, IN 47405, USA. [2] Present address: Solugen, Inc., 14549 Minetta St., Houston, TX 77035, USA. ✉email: jcl3@iu.edu

Halogen substituents can profoundly influence the biological activity of natural products, pharmaceuticals, and other organic compounds[1–4]. In nature, several classes of enzymes catalyze oxidative halogenation reactions that install halogen substituents (X = Cl, Br, I) into diverse structures starting from distinct precursors[5,6]. The high selectivity, mild reaction conditions, and environmentally benign oxidants and halogen sources used by these enzymes have motivated efforts to employ and engineer them for biocatalytic halogenation[7–9]. Flavin-dependent halogenases (FDHs)[10] have been particularly well-studied in this regard due to their ability to site-selectively halogenate electron-rich aromatic or enolate groups in a wide range of organic structures (Fig. 1a). These enzymes oxidize halide ions using $O_2$ as a terminal oxidant to generate a persistent lysine-derived haloamine that is believed to either directly halogenate substrates[11] or to serve as a source of HOX within the enzyme that can react with bound substrates[12]. Because many other oxidative halogenation reactions are initiated by similar electrophilic halogen species[13], we wondered whether FDHs might possess reactivity beyond aromatic halogenation, including enantioselective olefin halocyclization[14]. This reaction involves formal addition of a halenium ion ($X^+$) and a nucleophile ($Nu^−$) to an olefin, leading to the formation of two new bonds, adjacent stereogenic centers, and a halogen substituent that can be used for further structural elaboration (Fig. 1b). Relative to other FDH-catalyzed halogenation reactions, which involve only substitution of a C–H bond with a C–X bond[7,8], halocyclization is therefore notable for its ability to significantly increase molecular complexity.

Some vanadium haloperoxidases (VHPOs) that catalyze stereoselective halocyclization have been characterized, but only native or putative substrates containing stereogenic centers have been reported to date[16–18]. Other VHPOs[19,20] and heme haloperoxidases[21] can catalyze halocyclization of synthetic substrates, but racemic products are obtained, likely due to their formation via free HOX generated by the enzyme. No examples of FDH-catalyzed halocyclization or enantioselective halocyclization involving simple, achiral substrates using any enzyme have been reported. We reasoned, however, that FDHs could address several major challenges to enantioselective halocyclization by catalytically activating an inert stoichiometric halogen source[14] in a chiral environment[22] and prohibiting halenium ion transfer between olefins, which leads to racemization[23]. The chemical feasibility of halocyclization by a variety of N-halogenated amine species[14] and the broad substrate scope[7,8] and evolvability[24] of FDHs warranted exploration of this reaction manifold, which would significantly expand the catalytic repertoire of FDHs. In this study, we show that FDHs can indeed catalyze

enantioselective halocyclization of simple olefins and establish how active site mutations and mitigation of hypohalous acid release from the FDH active site lead to the high enantioselectivities observed.

## Results and discussion

A panel of FDHs comprising 45 wild-type enzymes from a recently reported genome mining effort[25] and 99 variants of the FDH RebH from previous directed evolution campaigns[26–29] was assembled (See supporting sequence data file). While halocyclization requires suitable orientation of an electrophilic halogen source, an olefin, and a pendent nucleophile and involves substrate deprotonation distal to rather than ipso to the site of halogenation as in aromatic halogenation (Fig. 1b), we anticipated that the diversity of enzymes in this panel would include examples that could accommodate these differences. FDH halocyclase activity was evaluated on 4-methoxyphenyl-4-pentenoic acid (1)[15]. We hypothesized that the styrene core of 1 could mimic the planar aromatic substrates accepted by many of the FDHs evaluated to orient the olefin for electrophilic attack within the FDH active site, and docking studies indicated that this orientation was indeed feasible.

Analysis of FDH activity on 1 using crude enzyme extract (see supporting information) led to the identification of 50 variants that provided significant yields of bromolactonization product 1a (Supplementary Fig. 1). These enzymes were purified, and the purified enzymes provided 1a in up to 94% assay yield and 84:16 e.r. (variant 4V + S[26]) using 5 mol% enzyme loading (Fig. 2a). In general, previously engineered variants of RebH displayed the highest selectivity, though several genome-mined FDHs provided a significant yield of the opposite product enantiomer (up to 36:64 e.r. with variant 1-F08[25]). In all cases, 5-exo-trig cyclization was observed, consistent with the strong electronic bias in 1 that favors this mode of cyclization (Fig. 1b)[15].

The reaction parameters were next modified to improve product yield and enantioselectivity (Fig. 2b). Early studies revealed that increasing substrate concentration substantially improved the enantioselectivity of some FDH variants (Supplementary Fig. 2). We hypothesized that a competing racemic halocyclization reaction involving HOBr was occurring outside of the FDH active site and that this reaction was suppressed at saturating substrate concentrations. Because the electrophilicity of HOBr decreases substantially above its pKa (8.7)[30], we reasoned that increased buffer pH would mitigate the reactivity of this species if it was released into solution. Indeed, the enantiomeric ratio of 1a produced by variant 4V + S increased from 86:14 (84% yield) to 96:4 (47% yield) as a result of using Tricine buffer in place of HEPES buffer and increasing pH from 7.4 to 9 (Fig. 2b, entries 1–4), albeit at the cost of product yield. Moreover, adding 1 mM glutathione, a known HOBr scavenger[31], improved enantioselectivity even at pH 7.5, providing 1a in 91% yield with 96:4 e.r. (Fig. 2b, entry 5). The optimized reaction conditions permitted halocyclization of 1 on 15 mg scale to provide 1a in 84% yield and 95:5 e.r. (Fig. 2b, entry 6). These results both suggest that free HOBr was being produced by 4V + S and highlight a means to eliminate the unwanted reactivity of this species to enable bromolactonization with high yield and selectivity.

The optimized reaction conditions were then used to evaluate FDH-catalyzed bromolactonization of six additional substrates (2–8, Fig. 3). A subset of purified variants from the initial screen using substrate 1 was examined to identify the best enzyme for each substrate (Supplementary Fig. 3). Good yields and high enantioselectivities were observed for halocyclization of substrates 4–7, which, like 1, contain an electron-rich aromatic moiety that would be expected to favor formation of the observed 5-exo

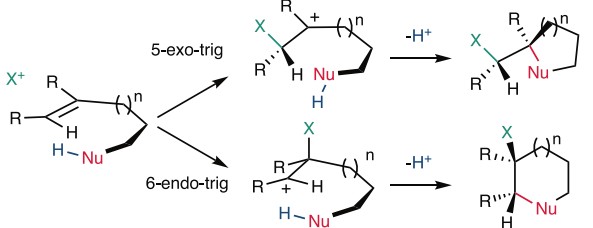

**Fig. 1 Expanding the scope of FDH catalysis. a** Arene/enol (R = OH) halogenation and **b** a simplified scheme[15] for olefin halocyclization.

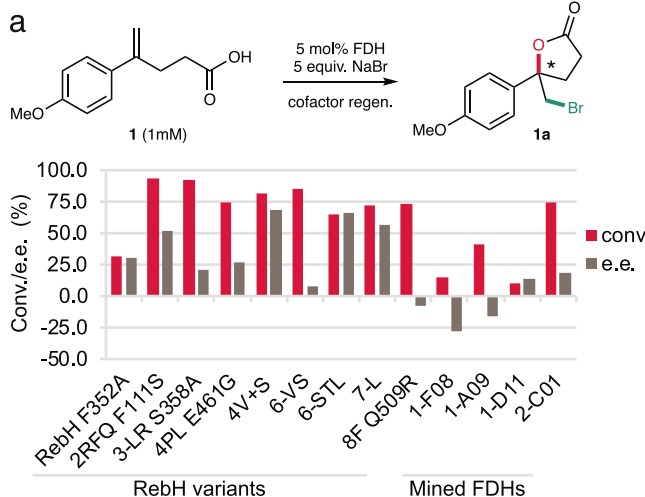

**a**

**b** optimization of reaction conditions for 4V+S

| Entry | pH (Buffer) | Glutathione | Yield (% 1a) | e.r. 1a |
|---|---|---|---|---|
| 1 | 7.4 (25 mM HEPES) | - | 84 | 86:14 |
| 2 | 7.6 (200 mM Tricine) | - | 83 | 90:10 |
| 3 | 8.5 (200 mM Tricine) | - | 77 | 92:8 |
| 4 | 9 (200 mM Tricine) | - | 47 | 96:4 |
| 5 | 7.6 (200 mM Tricine) | 1 mM | 91 | 96:4 |
| 6[a] | 7.6 (200 mM Tricine) | 1 mM | 84 | 95:5 |

**Fig. 2 Optimization of FDH-catalyzed bromolactonization. a** Selected yield and enantioselectivity data for bromolactonization of substrate **1**. **b** Optimization of halocyclization reaction conditions. Reaction mixtures contained 1 mM substrate, 5 equiv. NaBr, 5 mol% FDH, and a cofactor regeneration system comprising a flavin reductase, a glucose dehydrogenase, and glucose. Glutathione (1 mM), catalase, and optimized buffers were used as described in the supporting information. Product assay yields (hereafter, "yields") and selectivities are the average of triplicate measurements determined by LC/MS using *p*-bromoanisole internal standard. [a]Reaction conducted on 15 mg scale.

bromolactonization products. Substrates **4** and **5** possess additional bulk in their aromatic and aliphatic portions, respectively, while hexenoic acid derivative **6** shows that the FDH active site can accommodate formation of a 6-membered lactone. Finally, trisubstituted olefin substrate **7** (6.7:1 *E:Z*) underwent halocyclization in high yield to provide **7a** with modest diastereoselectivity (70:30) but high enantioselectivity (97:3 e.r. and 88:12 e.r., for the major and minor diastereomers, respectively, Supplementary Fig. 4), comparing favorably to the only report of a small molecule catalyst for this reaction[32]. While preliminary evaluation of chlorolactonization of **1** using a subset of the enzymes ultimately screened for bromolactonization revealed no hits, subsequent analysis of 4V + S revealed that it catalyzes chlorolactonization of **1** to give **1c** in 32% yield and 89:11 e.r. (Supplementary Fig. 5). The lower chlorolactonization yield is consistent with our previous observation that aromatic bromination of non-native substrates is generally more prevalent and higher yielding than the corresponding chlorination[25], and this promising activity bodes well for further directed evolution of FDHs with improved chlorolactonization activity.

Lower yields and enantioselectivities were observed for bromolactonization of electron-neutral (**2**, **8**) and electron-poor (**3**) substrates. This observation is consistent with a proposed difference in mechanism for the reaction of these substrates relative to **1**. Experimental and computational evidence suggests that bromolactonization of **2** proceeds via a concerted mechanism involving nucleophile-assisted alkene activation[15]. This would require proper *syn* or *anti* orientation of the active halogenating agent and the pendent carboxylate nucleophile relative to the olefin in a cleft of the FDH active site that natively binds planar aromatic substrates (Fig. 1). Bromolactonization of **1** appears to proceed via a stepwise mechanism[15], so precise arrangement of the nucleophile is not required. Electrophilic attack of the olefin by either a bromamine species[11] or HOBr[12] could generate a stabilized benzylic carbocation or halirenium intermediate that could be intercepted by the carboxylate following movement to a more accommodating orientation within the active site. It is also notable, however, that a mixture of regioisomers resulting from 5-exo-trig and 6-endo-trig bromolactonization (up to 84:16 r.r.), was obtained for the reaction of substrates **2** and **3**. While high regioselectivity was not obtained using the variants examined to

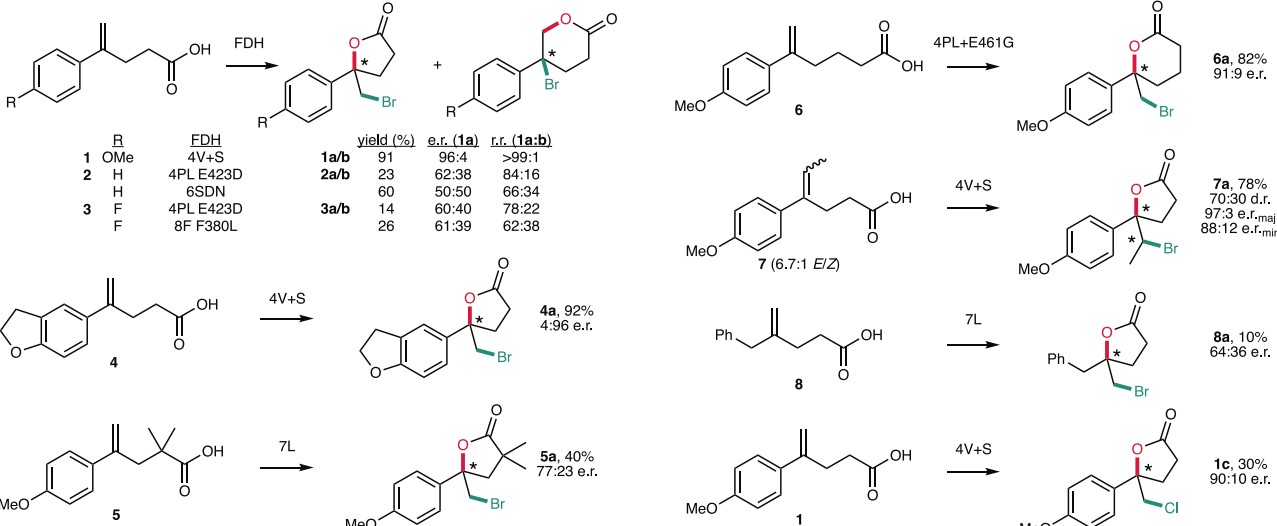

**Fig. 3 Halolactonization substrate scope of evolved RebH variants.** Reaction mixtures contained 1 mM substrate, 5 equiv. NaBr or 100 equiv. NaCl, 5 mol % FDH, and a cofactor regeneration system comprising a flavin reductase, a glucose dehydrogenase, and glucose. Glutathione (1 mM), catalase, and optimized buffers were used as described in the supporting information. Product yields and selectivities are the average of triplicate measurements determined by LC/MS using the internal standards indicated in the SI.

date, significant variation between variants was observed (66:34-84:16 r.r.), suggesting that the regioselectivity for halocyclization of electron-neutral or electron-deficient substrates could be controlled by a suitably engineered enzyme. Activity on benzyl-substituted substrate **8** also bodes well for the extension of halocyclase activity to olefins beyond styrenes. Aromatic halogenation was not observed for any of the substrates examined, consistent with the fact that FDHs typically only halogenate substrates with greater electronic activation than that imparted by one or two alkoxy substituents[33].

The above results show that FDHs can accommodate the significant differences between the substrates and intermediates involved in aromatic halogenation and olefin halocyclization (Fig. 1). Notably, however, all of the FDH variants that catalyze bromolactonization with high yields and selectivities (4V + S, 4PL E461G, and 7L) descend from RebH, which along with three other widely studied FDHs (PrnC, PltM, and ThHal)[7] provided <2% yield of **1a** at 5 mol% enzyme loading under optimized halocyclization conditions. The activity of variants along the lineages of the evolved enzymes was therefore examined to reveal mutations that improved halocyclase activity. In the case of 4V + S[26], for example, many mutations that improved yields for different aromatic halogenation reactions also improve halocyclization activity (Fig. 4a). Significant improvements in bromolactonization yield and enantioselectivity result from active site mutations N470S and F111S (Fig. 4a, entries 4 and 6), and 4V + S was the only variant in its evolutionary lineage that provided chlorolactonization activity (Supplementary Fig. 6). Variants 4V + S, 4PL E461G, and 7L possess in common active site mutations at only these residues, suggesting that they play particularly important roles in halocyclization catalysis.

Docking simulations were conducted to examine how F111S and N470S might affect FDH halocyclase activity. Several studies have established that substrates typically bind to FDHs such that the site of halogenation projects toward a conserved active site lysine residue (K79 in RebH)[24]. Geometry-optimized structures of **1** and the corresponding cationic brominated intermediate (Fig. 4b) were docked into the structure of RebH variant 3-LSR (PDB ID 4LU6), which possesses many of the mutations present in 4V + S, 4PL E461G, and 7L, but notably not F111S or N470S (PDB ID 4LU6). Low energy poses in each case show the substrate alkene (Supplementary Fig. 7a) or the intermediate bromine substituent (Fig. 4b) projecting toward K79. In the docked intermediate structure, the bromine substituent is only 3.4 Å from K79, but the propanoic acid substituent adopts an extended

conformation to avoid a steric clash with F111 that places the carboxylate 4.3 Å from the benzylic cation that it must attack for halocyclization to occur. Similar binding is observed using the N470S variant of 3-LSR (Supplementary Fig. 7c), so it is not clear how this mutation affects halocyclase activity.

The F111S mutation, on the other hand, allows the propanoic acid substituent to fold under both the intermediate (Fig. 4b) and the substrate structures (Supplementary Fig. 7b) in a conformation consistent with *syn* addition of carboxylate and Br⁺ to the olefin in a *pro-S* configuration. While low enantioselectivity for the *p*-fluorine-substituted substrate **3** was observed (Fig. 3), the major product had the *S*-configuration based on comparison to authentic enantio-enriched product, providing experimental support for the simulated binding poses (Supplementary Fig. 8). The bromine substituent in the intermediate structure remains 3.3 Å from K79, but the carboxylate is also only 3.3 Å from the benzylic site. This distance is 2.5 Å in the calculated transition state for chlorolactonization of the analogous substrate lacking the *p*-methoxy substituent (Supplementary Fig. 7c)[15]. This similarity suggests that the F111S/V mutations allow **1** to bind in orientation required for halocyclization that is otherwise blocked, which explains the large improvement in halocyclase activity and selectivity that results from F111S in the 4V + S lineage. F111 is highly conserved among FDHs[34], but we previously found that both F111S and F111V led to altered site selectivity for tryptamine halogenation[27], albeit with reduced activity, and similar changes for tryptophan halogenation by the homologous F94A mutant of PrnA were observed by Van Pée[35]. We later incorporated F111S into variant 4V and found that "4V + S" was among the most active enzymes evaluated for enantioselective desymmetrization of meso methylene dianilines (see Supplementary reference 29) suggesting that the larger active site cavity generated by F111S may be generally beneficial for reactions involving non-planar substrates and intermediates.

Finally, steady-state kinetics were used to compare FDH halocylase activity to the native aromatic halogenation activity of these enzymes. Variant 4V + S catalyzes bromolactonization of **1** with a $k_{cat}$ of 0.36 min⁻¹ and a $K_M$ of 0.47 mM (Supplementary Fig. 9). The similarity of these values to those for native FDH catalysis ($k_{cat} \sim 0.5$–3 min⁻¹)[24] highlight the facility with which the FDH active site can be reconfigured to enable topologically distinct chemical transformations. Previous studies of FDH-catalyzed aromatic chlorination established that mutating a conserved active site lysine (K79 in RebH) abolishes activity, and this was also observed for bromolactonization of **1** catalyzed by

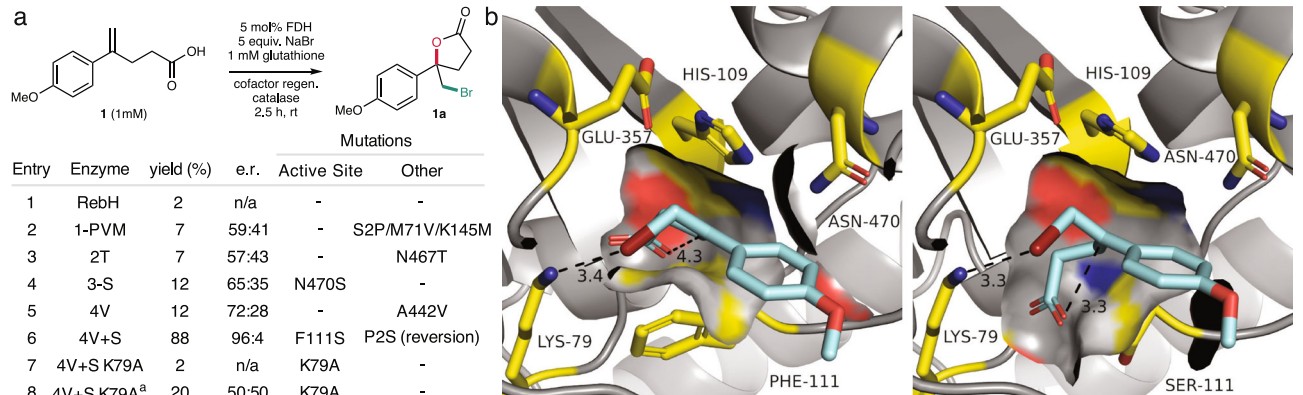

**Fig. 4 Structural origins of improved halocyclase activity in evolved FDHs. a** Effects of mutations in evolved FDHs on halocyclase activity. Mutations are listed relative to the parent in the previous row. Yields and selectivities are the average of duplicate measurements determined by LC/MS using *p*-bromoanisole internal standard. **b** Docking poses for the cationic intermediate generated upon bromination of **1** in the structure of RebH variant 3-LSR (left) and 3-LSR F111S (right). Key active site residues, including K79 and F/S111 are shown in yellow and a surface rendering of several residues is provided to illustrate the space created by F111S. The K79ε-amino-Br and Cbenzyl-Ocarboxylate distances are shown. ªReaction conducted without added glutathione.

| Entry | Enzyme | yield (%) | e.r. | Active Site | Other |
|---|---|---|---|---|---|
| 1 | RebH | 2 | n/a | - | - |
| 2 | 1-PVM | 7 | 59:41 | - | S2P/M71V/K145M |
| 3 | 2T | 7 | 57:43 | - | N467T |
| 4 | 3-S | 12 | 65:35 | N470S | - |
| 5 | 4V | 12 | 72:28 | - | A442V |
| 6 | 4V+S | 88 | 96:4 | F111S | P2S (reversion) |
| 7 | 4V+S K79A | 2 | n/a | K79A | - |
| 8 | 4V+S K79Aª | 20 | 50:50 | K79A | - |

4V + S K79A (Fig. 4a, Entry 7; Supplementary Figs. 10–12). In the absence of glutathione, however, bromolactonization activity for the K79A mutant was restored, but the reaction produced racemic product and was significantly slower than that of 4V + S in the presence of glutathione (Fig. 4a, Entry 8). No reaction was observed in the absence of FDH, indicting that this reaction did not result from side reactions involving reduced flavin cofactor, O₂, and halide ions in solution. Combined with our finding that the selectivity of less active enzymes like 3-LR can be increased by increasing substrate concentration (Supplementary Fig. 2), these results suggest that FDHs can release HOBr into solution even after it migrates into the active site[24]. Indeed, we confirmed that two electron-rich aromatic substrates could be brominated using RebH K79A (Supplementary Fig. 13), consistent with a recent study that showed that freely diffusible HOBr can be released from the active site of the FDH Thal[36]. Release of HOBr is exaggerated in the absence of K79, but this process can be mitigated by tuning the active site so that olefin substrates react before HOBr is released, as in 4V + S, or by reducing the reactivity of HOBr in solution using increased pH or added glutathione (Supplementary Fig. 14).

In summary, we have characterized the first examples of olefin halocyclization catalyzed by an FDH and the first examples of enantioselective halocyclization on simple, achiral substrates catalyzed by any enzyme. High yields and enantioselectivity could be achieved using enzymes previously engineered for arene halogenation, though notably not using a number of wild-type FDHs. While halocyclization is a key step in the biosynthesis of several natural products, all examples reported to date are catalyzed by VHPOs[18]. Our results indicate that FDHs should also be considered as viable catalysts for halocyclization in natural product biosynthesis. Given that many natural products contain aliphatic carbocycles and heterocycles bearing halogen substituents in orientations consistent with installation via halocyclization[4], it will be interesting to see if this possibility has been exploited in nature. Uncovering this manifold of FDH reactivity also led to the finding that RebH and possibly other FDHs can release freely diffusible HOBr from their active sites, indicating that caution is warranted when evaluating the activity of FDHs toward non-native substrates that may be reactive toward this species. This side reaction can be mitigated by controlling reaction pH or including reducing agents like glutathione, however, paving the way for exploration of other non-native transformations. Given the range of different halocyclization reactions and other organic transformations that proceed via oxidative halogenation[13], this expansion of FDH catalytic activity bodes well for the development of a wide range of biocatalytic halogenation reactions.

## Methods

**Materials**. Skirted 96-well PCR plates (product number 82006-704) were purchased from VWR International (Radnor, PA). Eppendorf un-skirted 96-well PCR plates (product number 951020362) were purchased from Fisher Scientific. Greiner Bio-One polypropylene 96-well V-bottom plates (product number 651201) were purchased from Fisher Scientific. Agilent 0.2 μm PVDF 96-well filter plates (product number 203980-100) were purchased from Agilent. Dialysis tubing (32 mm width; MWCO 6000–8000) was purchased from Fisher Scientific.

NAD, FAD, and antibiotics were purchased from Chem-Impex International Inc. (Wood Dale, IL). Antibiotics were prepared as 1000x stock solutions: 1000x chloramphenicol was prepared at 25 mg/mL in EtOH, and 1000x kanamycin was prepared at 50 mg/mL. Substrates were purchased from Sigma-Aldrich, Chem-Impex, AK Scientific, Enamine, and Santa Cruz Biotechnologies.

GDH-105 (hereafter, GDH; 50 U/mg) was obtained from Codexis, Inc. (Redwood City, CA). Catalase from the bovine liver was obtained from Millipore Sigma (2000–5000 U/mg; stock solutions were prepared assuming 2000 U/mg; product number C9322). Luria broth (LB) and Terrific broth (TB) media were purchased from Research Products International (Mt. Prospect, IL). Qiagen Miniprep Kits were purchased from QIAGEN Inc. (Valencia, CA) and used according to the manufacturer's instructions. Protein ladder (Blue Pre-stained

Protein Standard, Broad Range (11–190 kDa); product number P7706) was purchased from New England Biolabs (Ipswitch, MA).

Stock solutions of 10 mM NAD and 10 mM FAD were prepared in 25 mM HEPES pH 7.4 buffer (reaction buffer) and stored at −20 °C until use. Stock solutions of 1.5 M NaCl, 1.5 M NaBr, and 1 M glucose were prepared in reaction buffer and stored at 4 °C until use. Stock solutions of substrate were prepared at 100 mM in DMSO; for the high-throughput screen, substrate stocks were then diluted to 1 mM in reaction buffer, manually arrayed into 96-well plates as described later. RebF was expressed in *E. coli* BL21(DE3) as an MBP fusion from pLIC-MBP and stored at 140 μM in reaction buffer with 10% glycerol at −78 °C. GDH was prepared as 180 U/mL stock solution in reaction buffer immediately before reaction setup. For high-throughput screening, stock solutions of FDH were stored in reaction buffer with 10% glycerol and arrayed in 96-well un-skirted PCR plates as described later. Catalase stock solutions were prepared at 3500 U/mL in reaction buffer immediately before reaction setup.

**Instrumentation**. Thermal plate sealing with aluminum foil was performed using a Packard MicroMate 496 manual plate sealer. Measurement of DNA/protein concentration was performed using a Tecan Infinite 200 PRO plate reader on a Tecan NanoQuant plate. High-throughput LC-ESI-MS analysis was performed using an Agilent 1290 system equipped with an Agilent Eclipse Plus C18 column. CHIRAL column information. Chiralpak IH-U (3 mm × 50 mm), Chiralpak IG-U (3 mm × 50 mm), Chiralpak AD-H (4.6 mm × 250 mm). Preparative-scale bioconversions were purified using either: An Agilent 1100 HPLC equipped with a Supelco Discovery C18 semi-preparative column (25 cm × 10 mm, 5 μm particle size) and an Agilent 1260 Infinity II fraction collector using 0.1% formic acid in H₂O as the A solvent and 0.1% formic acid in acetonitrile as the B solvent.

**Software**. HPLC traces were processed using Agilent Chemstation Rev. C.01.08 (224). NMR spectra were processed using MestReNova 11.0. Plots were generated using GraphPad Prism 7.0 and Microsoft Excel (version 2013). Geometry optimization was conducted using Spartan 18, Wavefunction Inc., Irvine CA. Docking simulations were conducted using Autodock Vina as implemented in UCSF Chimera candidate version 1.13.1.

**Cloning and mutagenesis**. An alanine mutation was introduced into 4V + S gene by quick-change PCR. The PCR conditions were as follows: ~200 ng parent template, 0.3 μM forward primer, 0.3 μM reverse primer, molecular biology grade water, and 1X Primestar Max. Nucleotide sequences for all primers are given in Supplementary section II A (see Supplementary Information). PCR was performed in a volume of 50 μL with the following procedure: 98 °C 10 s, (98 °C 10 s, 62 °C 15 s, 72 °C 220 s) for 16 cycles, 72 °C 5 min. The resulting 4 V + S + K79A plasmid was purified and eluted with 35 μL hot water. This product was then subjected to DPN1(1 μL) digestion in cut smart buffer at 37 °C for 1 h. The plasmid was then gel purified using a Zymo DNA purification kit and transformed into BL21(DE3) *E. coli* cells. Transformed cells were recovered using 750 μL SOC medium for 1 h at 37 °C. Cells were spread on LB kanamycin agar plates and incubated at 37 °C overnight. Single colonies were picked, cultured, and stored as glycerol stocks; the desired mutations were verified by Sanger sequencing.

**Protein expression and purification**. Overall, 14 mL culture tubes containing 5 mL LB with required antibiotics were inoculated with a glycerol stock of BL21 (DE3) + pGro7 + pET28b(FDH) and incubated overnight at 37 °C, in the horizontal shaker at 250 rpm. The overnight culture (1 mL) was transferred into 100 mL TB in 500 mL conical flasks. The inoculated expression cultures were incubated at 37 °C and 220 rpm until the OD₆₀₀ reached ~0.6–0.8. At this point the incubator was cooled to 30 °C, was induced with 2 mg/mL L-arabinose and 100 μM IPTG final concentration. After 20 h expression, the cells were harvested by centrifugation at 2400 × g at 4 °C for 15 min. The supernatant media was discarded, and the cell pellets were resuspended in 15 mL buffer, pH 7.6. and disrupted by sonication on ice cold water. Cell lysates were clarified by centrifuging at 24,000 × g for 40 min at 4 °C. The soluble fraction of the lysate was decanted into a new 50 mL centrifuge tube, then transferred to 10 mL polypropylene frit-bottomed spin columns capped at the bottom and containing 500 μL Ni-NTA resin pre-equilibrated with equilibration buffer (20 mM phosphate, 300 mM NaCl, 10 mM imidazole, pH 7.6). Protein was bound to resin by gentle mechanical inversion of these centrifuge tubes for 1 h at 4 °C. After the binding step, the Ni-NTA suspensions were allowed to drain by gravity into a waste basin. Overall, 10 mL of wash buffer (20 mM phosphate, 300 mM NaCl, 25 mM imidazole, pH 7.6) was added to the columns, which were allowed to drain by gravity into a waste basin. The spin columns were nested within new 50 mL centrifuge tubes, and 10 mL elution buffer (20 mM phosphate, 300 mM NaCl, 250 mM imidazole, pH 7.6) was added and allowed to drain into the centrifuge tubes by gravity. The eluted protein solutions were transferred to 4 mL Amicon Ultra 10 K MWCO spin filters and concentrated by centrifugation at 4000 × g for ≈15 min at 4 °C. The protein solution was diluted with buffer, pH 7.6, and centrifuged again. Buffer exchange in this manner was performed 3–5 times, after which glycerol was added for a final concentration of 10% v/v. Protein concentration measured using absorbance at 280 nm using a Tecan NanoQuant plate with protein extinction coefficients.

**Small scale FDH reactions**. A solution of substrate (100 mM in DMSO) and FDH solution were added to a flat bottom 96 DWP containing the various reagents required for this reaction. The final concentrations of the reagents were as follows: 1 mM substrate, 20 mM glucose, 10 μM NAD, 10 μM FAD, 9 U/mL GDH, 35 U/mL catalase, 2.5 μM MBP-RebF, and 50 μM FDH. The resulting mixture was left shaking at 650 r.p.m. at RT for varying amount of time. After shaking the reaction at rt the reactions were quenched, and proteins precipitated by adding 70 μL MeOH premixed with appropriate internal standards was centrifuged at $2400 \times g$, 4 °C for 5 min. Using a multichannel pipette, 70 μL of supernatant from the reaction plates were transferred to new 96-well filter plate retaining the well layout of the previous plate. The filter plate with a new 96 V-bottom deep well plates was then centrifuged at $2400 \times g$, 4 °C for 5 min. The collected filtrate was then sealed with aluminum foil and analyzed using LCMS.

**Large scale FDH reactions**. Substrate, small molecule, 4V + S, and a cofactor regeneration mixture were added in a 500 ml Erlenmeyer flask allowed to shake at 140 rpm for 18 hours. The reaction mixture was transferred to a connical tube and protein was pelleted by centrifugation at $24,000 \times g$ at 4 °C for 30 min. Supernatant was transferred into a separatory funnel and extracted with ethyl acetate five times. The precipitated protein was then resuspended in methanol, centrifuged at $24,000 \times g$ at 4 °C for 30 min. The organic layers were combined, dried over sodium sulfate, concentrated, and purified using semi-preparative HPLC.

**Geometry optimization**. Geometry optimization of substrate 1 (deprotonated carboxylate) and the corresponding cationic $Br^+$ adduct (deprotonated carboxylate) were conducted using Spartan 18, Wavefunction Inc., Irvine CA. The equilibrium geometry at ground state in polar solvent using the appropriate charge state was calculated using DFT with the B3LYP functional and a 6-31G* basis set. Coordinates of the optimized structures follow are provided in the Supplementary information.

**Docking simulations**. Docking simulations were conducted using Autodock Vina as implemented in UCSF Chimera. Geometry-optimized ligands were first imported into Chimera and prepped with Dock Prep using the default program settings, other than the appropriate charge. The crystal structure for RebH variant 3-LSR (PDB ID 4LU6) was prepped with Dock Prep using the default program settings. Mutants of this enzyme were generated in Chimera using the Rotamers application; the highest probability rotamer was used. The appropriate prepped structures were loaded into Autodock Vina, and the receptor search volume was set to encompass the entire FDH active site. Selected models were exported as .pdb files for visualization and graphic preparation in Pymol (see Supplementary Figures).

**Steady-state kinetics parameters for 4V + S bromolactonization**. Determination of kinetic parameters: rates were determined by monitoring the conversion of substrate 1 in the presence of NAD (100 μM final concentration), FAD (100 μM final concentration), NaBr (5 eq), MBP-RebF (2.5 μM final concentration), glucose dehydrogenase (9 U/mL final concentration GDH), glucose (20 mM final concentration), and p-bromoanisole as an internal standard (0.1 mM final concentration) at a final volume of 70 μL in a microtiter plate. 4V + S was added at a final concentration of either 20 μM. Plates were sealed using a plate sealer and shaken at 650 rpm at room temperature. Reaction mixtures were quenched by addition of 70 μL of MeOH. The precipitated protein was then removed by centrifugation and the reactions were filtered and analyzed by UPLC. Product formation was determined by calculating the ratio of product to internal standard and fitting that value to a calibration curve prepared from known concentrations of product 1a. The kinetic parameters ($K_M$ and $k_{cat}$) for 4V + S were determined from the substrate concentrations and the observed initial rates.

**General synthetic procedures**. Reactions for substrate and racemic standard preparation were prepared in flame or oven-dried glassware under an inert $N_2$ atmosphere. Thin-layer chromatography plates were visualized using 254-nm ultraviolet light. Flash column chromatography was carried out using Silicycle 230–400 mesh silica gel. $^1H$ and $^{13}C$ NMR spectra were recorded at 500 or 400 and 126 MHz, respectively, on a Varian spectrometer, and chemical shifts are reported relative to residual solvent peaks. Chemical shifts are reported in p.p.m. and coupling constants are reported in Hz. Yields for FDH-catalyzed reactions were determined by HPLC with respect to internal standards and reported as the average of two/three trials. Low-resolution ESI mass spectra were obtained using Agilent LC-MS. Full synthetic procedures and characterization are provided in the Supplementary information.

**Reporting summary**. Further information on research design is available in the Nature Research Reporting Summary linked to this article.

## Data availability

All methods and data supporting the findings of this study are available within the paper or its supplementary information files (Supporting information.docx). The structure of RebH variant 3-LSR is available at www.rcsb.org under PDB ID 4LU6. Source data are provided with this paper.

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

## Acknowledgements

This study was supported by the NIH (R01 GM115665) and a Camille and Henry Dreyfus Foundation Teacher Scholar Award. D.M. was supported by the Precision Health Initiative of Indiana University. B.F.F. was supported by an NIH Ruth L. Kirschstein National Research Service Award (F32 GM123693). Halogenase genes from reference [25] were provided by the US Department of Energy Joint Genome Institute, a DOE Office of Science User Facility, supported under contract no. DE-AC02-05CH11231. Results in this study were acquired using an NMR spectrometer funded by NSF MRI CHE-1920026 and a Prodigy probe was partially funded by the Indiana Clinical and Translational Sciences Institute. We thank Christian Gomez, Harrison Snodgrass, and Dr. Mary Andorfer for insightful discussions, Prof. Nicola Pohl for providing access to her laboratory space, and previous group members who evolved the enzymes used in this study.

## Author contributions

D.M. completed substrate synthesis, all biocatalysis screening, optimization, kinetics, and characterization. B.F.F. optimized the mass spectrometry assay for initial biocatalyst screening. Y.J. assisted with substrate synthesis and product characterization. J.C.L conceived and directed the project and completed the docking simulations.

## Competing interests

The authors declare no competing interests.
