## [Peer Review File · Nature Communications]

Reviewers' Comments:

Reviewer #1:

Remarks to the Author:

The manuscript by Mondal et al. reports new use of flavin-dependent halogenases (FDH) in catalyzing olefin halocyclization.

Although a report on this biocatalytic activity is interesting, the manuscript lacks rigorous characterizations and in-depth explanation of this activity of FDH.

1. Title, this work does not report any directed evolution work. The authors used existing enzymes in their library to explore.

2. Recently, Phintha et. al (J Biol Chem, 2020), reported that significant amount of HOBr can leak out of the wild-type FDH active site. In this manuscript, the authors also report that HOBr can leak out from the K79A variant active site. I wonder if the halocyclization detected with other enzymes is also caused by this free HOBr.

3. Following from the above comment, the key issue is whether the substrates investigated in this paper really bind to the enzyme's active site. The docking data only tell that the binding is possible but it does not mean they really bind.

-I suggest the authors do competition experiments in which both native substrates such as tryptophan are also added to compete. If the yield of olefin halogenated products is lowered in the presence of tryptophan, it would support that the olefin substrate can bind to the active site.

-Alternatively, native reactions (bromination and chlorination) of aromatic compounds should be tested with these enzymes. If these enzymes can bind to aromatic compounds tested, simple halogenated products should be observed.

4. As noted above about validity of the docking data, the binding configuration in Figure 2 is not convincing. From Figure 2B, the carboxylate group of compound 1 is positioned close to Glu-357. The interaction between two carboxylate groups should create repulsion.

5. As the reaction is also involved with cyclase activity in which compound conformational changes are required, the docking results alone cannot explain the cyclase activity where high dynamic occurs. The authors would obtain insightful data by performing MD simulations and comparing among variants, for instance, 4-V+S versus 4-V.

6. As for the substrate binding to 4-V+S which has F111S, the experimental evidence to support that this enzyme can bind to aromatic compounds is required. As this residue is important for substrate binding and highly conserved among FDHs (PrnA, PyrH, SttH, Thal), its replacement with serine may not accommodate binding of the substrate 1 or other aromatic compounds.

7. Figure 1C, how can different buffers affect enantiomeric selectivity of the reactions? These effects should be clarified.

8. Page 3, third paragraph, positive control experiments with tryptophan chlorination should be carried out to make sure that the same enzymes incapable to catalyzing chlorolactonization were active.

A previous report (Younes et al., ChemSusChem 2020) demonstrated that HOCl can be incorporated in similar lactonization reactions. The yield from using HOCl was similar or slightly lower than the HOBr-catalyzed reactions. Because RebH and other FDHs can catalyze chlorination with a similar rate to bromination (Yeh E. 2005), I do not understand why the authors did not observe chlorolactonization.

9. Page 6, effects of glutathione on activities of the wild-type and K79A should be explained.

10. Regarding HOBr scavenging activity, in reference no. 29, HOBr scavenging rate is not reported.

11. Description and basic information of each variant should be provided so that readers can analyze and understand mutational effects.

12. Were bioconversion experiments only performed in a single set, not in triplicate?

Reviewer #2:

Remarks to the Author:

The power of FDHs in regioselective aromatic halogenations, often even overriding the intrinsic position selectivity, has been demonstrated by huge body of work dealing with this interesting enzyme class. Nevertheless, halogenations catalysed by FDHs beyond aromatic substitution have been elusive. Now, Lewis and co-workers evolved the native RebH to address unnatural halogen-triggered transformations by directed evolution. The authors impressively showcased in the presented manuscript a chemo-enzymatic approach for high enantioselective halocyclization of olefins by using flavin-dependent halogenase variants. The 4V+S variant showed the most promising results and was further examined in detail. The lysine residue K79, F111S mutation and the prevention of freely diffusible HOBr proved to be essential to achieve good yields and high enantioselectivity. Unfortunately, the corresponding chlorolactonizations were not accomplished. Overall, the manuscript is a great piece of work highlighting the exciting chemistry of this enzyme class. The studies have been thoroughly conducted and the hypothesis drawn from the experimental results are overall conclusive. I only can congratulate the authors to such a great and comprehensive investigation, which I enjoyed to read. I fully recommend publication of the article.

Major Comments:

Comment 1 (regarding optimization of reaction conditions; figure 1C)

From the text it is not clear if the improved reaction conditions with glutathione as HOBr scavenger were tested with other variants than the 4V+S. What about other mutants with promising yields and good e.e.s? Is this concept transferable?

Minor Comments:

Comment 2 (regarding chlorolactonization; line 96-98)

It would be nice to go into more detail, what that previous work (citation 23) showed about the halogen specificity. What could be the reason for the missing chlorolactonization activity on a molecular level? Do you think the main reasons is the distance of the carboxylate to the benzylic site?

Comment 3 (regarding unspecific reaction with HOX)

It is assumed that there is freely diffusible HOX released by the enzyme, which leads to an unspecific reaction outside of the active site and produces a racemic mixture. It would be preferable to write a few more sentences about free HOCl. Can it also be released? Why was no racemic mixture of the chlorolactonization product detected?

Tanja Gulder

Reviewer #3:

Remarks to the Author:

The manuscript entitled "Directed Evolution of Flavin-Dependent Halogenases Enables Non-Native Enantioselective Olefin Halocyclization Catalysis" submitted by Lewis and co-authors reports on the development of halogenating enzymes towards non-native catalysis. As the authors point out, halocyclization is an interesting and important transformation in synthetic chemistry, whilst the introduction of stereoselectivity provides a challenge. In the field of biocatalysis, this reaction type has only been described for haloperoxidases which are less attractive due to their non-specific release of hypohalous acid. The authors' approach to introduce halocyclization into flavin-dependent halogenases (FDHs) is therefore very interesting and unprecedented. FDHs are equipped with a remarkable selectivity allowing regioselective halogenation of electron-rich arenes.

Despite their attractive selectivity and immense investigations FDHs have been shown to suffer from severe drawbacks, mainly low efficiency, whereas the field of non-native catalysis has been overlooked so far. Therefore, the present manuscript covering enantioselective halocyclization is highly innovative and a significant step forward.

The manuscript is well written, experiments and resultant data were thoroughly explained. Starting from a model olefin the authors screen a library of in-house FDHs towards the non-native activity, thus giving a reduced set of positive hits with varying conversion and enantioselectivity. Interestingly, the engineered variants within the screen turned out to be more active than the wild type catalysts. Well-conceived optimisation finally resulted in improved selectivity and conversions so that bromolactonization was well-established and is successfully demonstrated for a small array of olefins on an analytical scale. The non-specific release of HOBr by the FDH has not been reported before and turned out to be most decisive to control stereoselectivity. By a smart variation of reaction parameters and substrate loading the authors were able to overcome this issue and abolish non-specific bromolactonization. The findings were complemented by docking studies which provide first insights into binding modes and how active site mutations might favour halocyclization within the FDH.

Before publication I suggest to resolve the following issues:

1. Manuscript title: The title says "directed evolution" which implies that the paper includes an engineering study. Yet I cannot find any protein engineering performed in this study. If I am not mistaken, the enzyme variants reported here (apart from the control mutant K79A) come from a previously established panel, mainly a substrate walking approach previously published by the group (ref. 24) as well as genome mining (ref. 23). If engineering efforts contributing to improved halocyclization activity were performed herein, this must be mentioned more clearly in the manuscript (and also added to the experimental section). Otherwise I suggest considering a different title.

2. The authors use the term 'yield' when referring to biotransformation efficiency. Do they refer to 'isolated yields' or 'conversions' based on HPLC traces? Yield usually implies that a reaction product is isolated (at least on a milligram scale), purified and structurally characterised, e.g. by NMR. I suppose this has not been done here as the biotransformations were carried out on a scale of 70 μ L. Thus, 'conversion' seems to be more appropriate to me as the compounds have not been isolated. I recommend revising this throughout the manuscript.

3. One of the most interesting achievements is that reaction optimisation successfully improves the enantioselectivity of halocyclization by quenching HOBr through a change of pH, glutathione addition, or increased substrate loading. This is particularly astonishing since for FDHs (unlike haloperoxidases) the release of HOBr from the active site has not been observed before. The authors claim that this suggests a finding to be taken into account for future investigations on FDHs, e.g., when expanding the substrate scope (conclusion, lines 227-230). I understand that this is an important statement requiring further evidence, particularly because it is one of the key findings towards higher reaction selectivity in this study. Therefore, the authors could provide further data if the non-specific release of HOBr also impacts 'native halogenation', e.g., of indole(s), anilines, or if it is a side effect merely important for non-native catalysis. Moreover, the authors' data suggest that the K79A mutant is able to release HOBr which is reasonable as the halogenating species is formed in the flavin-binding site and does not involve K79. I wonder if this mutant leads to detectable (trace) halogenation as well. These interesting observations need further clarification prior publication because they closely relate to some basic investigations on halogenase catalysis by the groups of van Pee and Walsh.

4. The manuscript requires more details on how biotransformations were analysed. I assume RP-HPLC(-MS) was utilised to analyse the enzyme panels. How were the products confirmed? Did the authors compare HPLC traces with authentic standards? The Supporting Information contains synthetic details on substrate synthesis. Were the (racemic) lactone products synthesised as well?

5. It would be a good improvement if the authors could comment on absolute configurations in addition to enantiomeric ratios where possible, although I understand that this is hard to decipher. For clarity to the readers, please indicate chiral centres with an asterisk.

6. Does background halogenation activity occur as a competing reaction, for instance, on the benzene moiety? As the halogenated products and the halolactone have got an identical mass further commenting is necessary. Also HPLC diagrams showing product formation could be added.

7. To complete the story, it is important to run representative transformations on a low milligram scale. In this case, the purified products can be structurally characterised (NMR, HR-MS etc.) to finally confirm bromolactonization. I am aware that running larger-scale reactions is challenging using halogenases. However in light of the good conversions achieved for substrate 1 using mutant 4V+S, for instance, I am confident that this will be feasible on a scale of 5 mg. Moreover, this will further corroborate the applicability of halocyclization in synthesis of complex small molecules which will strengthen the impact of the manuscript.

Overall, this manuscript is self-contained and of high novelty. Experiments were carried out thoroughly and conclusions drawn are convincing. The text is clearly written and the data are presented well (please note my comments on further edits below). Given the minor alterations suggested above are addressed the paper merits publication in Nature Communications.

Further edits:

- Figure 1 contains too many details from different experiments. I suggest splitting the figure into 2-3 individual figures / tables and place them in the text where mentioned. For example: (A) // (B+D) // (C). Also the table in Figure (D) should be enlarged.
- Figure 2B: Residue S111 is hardly visible.
- References: The authors should add more references from other groups who contributed to enzymatic halogenation in the past years.
- line 89-98: Please comment on the way of analysis. Adding an HPLC diagram, e.g., a time course (start + end point) might be a suitable illustration.
- line 109: "pH from 7.4 to 9 (Fig. 1C, entries 1-4), albeit at the cost of product yield." – Does saponification of the lactone occur at pH 9?
- line 136: "Electrophilic attack of the olefin by a bromamine species (...)" – as the authors know it is ambiguous if the bromoamine actually acts as the halogenating species (Flecks et al. DOI: 10.1002/anie.200802466). Most likely HOBr performs the attack. Probably this is also the case for halocyclization.
- line 140: 'was obtained' ; refers to 'mixture' (singular)
- line 157: "and variants 4V+S, 4PL E461G, and 7L possess in common active site mutations at only these residues" – E461G is also located in/ very close to the active site as it coordinates Trp-NH2 in the native reaction.
- S5, line 177: spelling 'plate' / 'was'
- S6, line 197: spelling 'reagent'
- S6, line 202: spelling 'plate' / 'was'
- S23, line 582: please correct product of Wittig reaction to alkene
- S23, line 590: 'substituted' (no capital letter)
- S24, line 610: Please correct the reaction product (alkene)

REVIEWER COMMENTS

Reviewer #1 (Remarks to the Author):

1. Title, this work does not report any directed evolution work. The authors used existing enzymes in their library to explore.

The original title was used to emphasize that only enzymes obtained from directed evolution, rather than naturally occurring enzymes, catalyzed halocyclization with any significant activity or enantioselectivity. Given that reviewer 3 brought this issue up also, however, we changed the title to “Flavin-Dependent Halogenases Catalyze Enantioselective Halocyclization”.

2. Recently, Phintha et. al (J Biol Chem, 2020), reported that significant amount of HOBr can leak out of the wild-type FDH active site. In this manuscript, the authors also report that HOBr can leak out from the K79A variant active site. I wonder if the halocyclization detected with other enzymes is also caused by this free HOBr.

Enzymatic halocyclization has only been previously reported using V- and heme-dependent haloperoxidases. In cases where simple, achiral substrates are used, racemic products are obtained, and this is believed to result from free HOBr as the reviewer suggests. This was noted in the manuscript: “Other VHPOs^{19,20} and heme haloperoxidases (HHPOs)²¹ can catalyze halocyclization of synthetic substrates, but racemic products are obtained, likely due to their formation via free HOX generated by the enzyme.”

3. Following from the above comment, the key issue is whether the substrates investigated in this paper really bind to the enzyme’s active site. The docking data only tell that the binding is possible but it does not mean they really bind.

-I suggest the authors do competition experiments in which both native substrates such as tryptophan are also added to compete. If the yield of olefin halogenated products is lowered in the presence of tryptophan, it would support that the olefin substrate can bind to the active site.

-Alternatively, native reactions (bromination and chlorination) of aromatic compounds should be tested with these enzymes. If these enzymes can bind to aromatic compounds tested, simple halogenated products should be observed.

The high enantioselectivity observed as a result of active site mutations is strong evidence in support of catalysis occurring in the enzyme active site. Docking was not (and I believe cannot) be used as evidence to support whether a ligand will bind into an enzyme in solution—it was only meant to illustrate how substrates might be oriented in the active site if they did bind. The aromatic halogenation activity of the enzymes in Figure 4A, entries 1-5 were reported in reference 26; they were originally evolved for improved activity on aromatic compounds. Nonetheless, we examined the activity of 4V+S in the presence and absence of a methylene dianiline substrate that 4V+S itself was original engineered to halogenate (see below for more on this). A comparison of the time course for these reactions are shown in the SI (Fig. S15), and this shows that the qualitative rate of halocyclization product formation is reduced as expected if both substrates are binding in the FDH active site.

4. As noted above about validity of the docking data, the binding configuration in Figure 2 is not convincing. From Figure 2B, the carboxylate group of compound 1 is positioned close to Glu-357. The interaction between two carboxylate groups should create repulsion.

I appreciate that it can be difficult to visualize 3D structures in 2D images. The distance between the closest oxygen atoms in E357 and the substrate carboxylate is 5.3 Å, so the electrostatic repulsion should be relatively weak. On the other hand, the closest oxygen atom of the substrate carboxylate is only 3 Å from K79, which would be an attractive interaction. The docking simulations were conducted as described in the SI and account for appropriate substrate charges and other intermolecular forces, which can be difficult to estimate by inspection.

5. As the reaction is also involved with cyclase activity in which compound conformational changes are required, the docking results alone cannot explain the cyclase activity where high dynamic occurs. The authors would obtain insightful data by performing MD simulations and comparing among variants, for instance, 4-V+S versus 4-V.

We agree. For this reason, we used the guarded statement that “Docking simulations were conducted to examine how F111S and N470S might affect FDH halocyclase activity.” We intend to pursue MD studies to better understand the unique reactivity revealed in the current manuscript, but that will also require substantial experimental mechanistic analysis, including steady state kinetics for all relevant variants, that is beyond the scope of this initial study.

6. As for the substrate binding to 4-V+S which has F111S, the experimental evidence to support that this enzyme can bind to aromatic compounds is required. As this residue is important for substrate binding and highly conserved among FDHs (PrnA, PyrH, SttH, Thal), its replacement with serine may not accommodate binding of the substrate 1 or other aromatic compounds.

We first reported the activity of an FDH containing F111S and F111V in reference 27 (see variants 5LS and 6S); they led to altered site selectivity in tryptamine halogenation. 4V+S was created during a later effort to engineer FDHs for enantioselective desymmetrization. As described in reference 29, it has good activity, but the enantioselectivity was too low to include in the manuscript, so the activity data ended up only in the SI. To address the reviewer’s concern, we added the following text to the manuscript:

F111 is highly conserved among FDHs,³⁴ but we previously found that both F111S and F111V led to altered site selectivity for tryptamine halogenation,²⁷ albeit with reduced activity, and similar changes for tryptophan halogenation by the homologous F94A mutant of PrnA were observed by Van Pée³⁵. We later incorporated F111S into variant 4V and found that “4V+S” was among the most active enzymes evaluated for enantioselective desymmetrization of meso methylene dianilines (see SI for reference 27),²⁹ suggesting that the larger active site cavity generated by F111S may be generally beneficial for reactions involving non-planar substrates and intermediates.

7. Figure 1C, how can different buffers affect enantiomeric selectivity of the reactions? These effects should be clarified.

Our best guess is that the enzyme appears to be more stable in Tricine vs HEPES based on observation of increased precipitation of enzyme over time in HEPES buffer. This could lead to degradation of enantioselectivity over time in HEPES. The effect is small and this explanation is only speculative, so I don't think it would be appropriate to comment on the origin of the observed affect in the manuscript.

8. Page 3, third paragraph, positive control experiments with tryptophan chlorination should be carried out to make sure that the same enzymes incapable to catalyzing chlorolactonization were active.

The enzymes described in this paragraph came from the larger set described in the manuscript: "A panel of FDHs comprising 45 wild-type enzymes from a recently reported genome mining effort²⁵ and 99 variants of the FDH RebH from previous directed evolution campaigns²⁶⁻²⁹". Extensive characterization of aromatic chlorination reactivity for all of these enzymes was reported in the references cited. Nearly all of the evolved enzymes and many of the mined enzymes are active chlorinases, which addresses the reviewer's request.

A previous report (Younes et al., ChemSusChem 2020) demonstrated that HOCl can be incorporated in similar lactonization reactions. The yield from using HOCl was similar or slightly lower than the HOBr-catalyzed reactions. Because RebH and other FDHs can catalyze chlorination with a similar rate to bromination (Yeh E. 2005), I do not understand why the authors did not observe chlorolactonization.

The manuscript by Younes et al. (ref 19) deals with a V-dependent haloperoxidase that releases into solution free HOX (X = Cl, Br), which subsequently promotes racemic halocyclization at pH 5.5. This is different than FDH catalysis, which typically involves a unique active site halogenating agent. As stated in the manuscript: "These enzymes oxidize halide ions using O₂ as a terminal oxidant to generate a persistent lysine-derived haloamine that is believed to either directly halogenate substrates¹¹ or to serve as a source of HOX within the enzyme that can react with bound substrates¹²." If the active halogenating agent is anything other than aqueous HOX, which we believe is the case for FDHs in the presence of glutathione, then one would not expect to see the same reactivity as haloperoxidases, particularly given the difference in the pH values at which haloperoxidases and FDHs are used (more on this below).

That being said, this situation has become more complex as we sought to clarify the role of free HOX in FDH-catalyzed halocyclization in response to reviewer comments. Most notably, our initial claim that chlorolactonization was not observed was based on an early screen of a subset (18) of the enzymes ultimately screened for bromolactonization, but 4V+S was not among those because it had not yet been identified as a top hit for bromolactonization (see Fig. S5 for details). When we saw many enzymes in that subset that gave bromolactonization but none that gave chlorolactonization, we gave up on the latter, and focused our more extensive screening efforts on bromolactonization. This was a mistake. When we specifically examined chlorolactonization activity in the 4V+S lineage in response to the reviewers comment, we found that 4V+S *does*

catalyze this reaction to give a 32% yield and ~89:11 e.r. No other enzymes in the lineage possess chlorolactonization activity, which further highlights the importance of the F111S mutation. We are grateful to the reviewers for prompting us to review our findings. The manuscript was modified in several places to explain this situation. We deleted the initial note that chlorolactonization was not observed, added an explanation of chlorolactonization evaluation, and added notes about chlorolactonization in the scope and lineage analysis sections.

9. Page 6, effects of glutathione on activities of the wild-type and K79A should be explained.

We have added an analysis of bromo- and chlorolactonization by both wt RebH and 4V+S and the corresponding K79A mutants (Figures S10-S12) to the SI to address the reviewers request, and these are called out in the manuscript. That analysis and some additional context is summarized here, and the key points are included in the SI:

The bromolactonization results mirror what was reported in the original manuscript. 4V+S provides optimal yield and enantioselectivity in the presence of glutathione. The K79A mutant of this enzyme provides modest yield of essentially racemic product in the absence of glutathione, which we ascribed to reaction by free HOBr. RebH and its K79A variant provide only trace product in the presence of glutathione, but a similar yield of racemic product (~20%) is observed for the K79A variants of both 4V+S and RebH in the absence of glutathione. In most cases, slightly higher yields are observed for the K79A mutants at pH 6.8 relative to 7.6, which is consistent with the [HOBr]/[BrO⁻] ratio expected at the two pH values (79:1 and 13:1).

Our initial manuscript only reported bromolactonization using 4V+S K79A results since, as noted above, we had not observed chlorolactonization using any of the variants investigated when those experiments were conducted. We were surprised to find that we do not observe chlorolactonization using 4V+S K79A (or RebH K79A) in the absence of glutathione at pH 7.6 or 6.8. To check for reactivity at the lower pH used by Younes et. al. (5.5), where FDHs are not stable, we also incubated the K79A variants under normal reaction conditions in the absence of substrate, filtered the enzyme, adjusted the pH, and added substrate, but no product was observed. This procedure mirrors that used in reference 36 to look for aromatic bromination by free HOBr, but it should be noted that very low levels of brominated product and no chlorinated product were detected in that study. We had attempted similar approaches to generate free HOX for subsequent assay, but we never saw anything convincing. It is possible that this approach is not generally useful for accurately determining “leaked” HOX.

The bromolactonization results for 4V+S and its K79A variant in the absence of glutathione show that the active brominating species in FDH catalysis provides greater conversion than HOBr released by the enzyme. The bromolactonization results for 4V+S in the presence and absence of glutathione show that the small amount of racemic product proposed to result from HOBr released from the enzyme (with K79 present) in the absence of glutathione is sufficient to significantly erode product enantioselectivity. In all cases, slightly higher yields are obtained for 4V+S in the absence of glutathione versus in the presence of glutathione. Similar trends in chlorolactonization yields and selectivities using 4V+S are observed +/- glutathione.

While one might expect the proposed racemic background reaction to be exaggerated with the K79A mutants, the bromolactonization results show that this is not necessarily the case. RebH

K79A indeed provides a significantly higher yield of racemic bromolactonization product (18%) relative to the trace observed with the wt enzyme in the absence of glutathione. On the other hand, significantly higher bromolactonization yield is observed for 4V+S relative to 4V+S K79A (86% vs 17%) in the absence of glutathione at pH 7.6 (similar at pH 6.8). We suspect that this difference reflects the significantly improved ability of the 4V+S active site to accommodate halocyclization relative to RebH.

4V+S only provides 35% yield for chlorolactonization in the absence of glutathione. A similar reduction in yield for the corresponding K79A variant as observed for bromolactonization (80%) would lead to a chlorolactonization yield of around 7%. As noted above, however, the [HOX]/[XO⁻] ratio is ~15 times higher for X=Br than for X=Cl at the pH values used for “live” reactions, which could explain the lack of racemic background reaction observed for the latter. As noted above, it is unclear that lowering the pH of the reaction mixture after an incubation period is a reasonable way to assay for generation of HOCl at pH values that would give a large [HOCl]/[ClO⁻] ratio. It is also not clear why chlorolactonization enantioselectivity decreases in the absence of glutathione since there does not seem to be sufficient racemic background reaction based on the 4V+S K79A results to account for the observed decrease. If HOCl leaked from 4V+S leads to reduced enantioselectivity from a racemic background reaction in the absence of glutathione, why don't we see any racemic reaction for 4V+S K79A? I suspect that the very different lifetimes of chloramines and bromamines (see ref 11 for discussion) is involved, but these details will require significant mechanistic studies to resolve, and they are beyond the scope of this initial disclosure.

10. Regarding HOBr scavenging activity, in reference no. 29, HOBr scavenging rate is not reported.

Reference 29 (now 31) is cited to clarify the role of glutathione (“glutathione, a known HOBr scavenger,³¹”), not the rate of the reaction between glutathione and HOBr, so I don't think this is needed. I did a little searching, and I can't find a specific reference that measures the rate of reaction between HOBr and glutathione (rates for HOCl are common due its biological relevance). If the reviewer has one in mind, I am happy to include it.

That being said, Reference 31 states that “Competition kinetic studies show that the selenospecies react with HOCl with rate constants in the range $0.8\text{--}1.0 \times 10^8 \text{ M}^{-1}\text{s}^{-1}$, *only marginally slower than those for the most susceptible biological targets including the endogenous antioxidant, glutathione.* [...] *Rate constants for reaction of the seleno-sugars with HOBr are ~8 times lower than those for HOCl ($1.0\text{--}1.5 \times 10^7 \text{ M}^{-1} \text{ s}^{-1}$).*” This suggests that the rate for HOBr scavenging for glutathione should be around $10^7 \text{ M}^{-1} \text{ s}^{-1}$ —a very fast reaction even if this estimate is off by a couple orders of magnitude.

11. Description and basic information of each variant should be provided so that readers can analyze and understand mutational effects.

A full list of all variants, names, and mutations was provided in the SI. See the “Screening Data” file on the NCOMMS website. This has been updated to more clearly show mutations and Uniprot IDs where relevant.

12. Were bioconversion experiments only performed in a single set, not in triplicate?

Initial screens were only single measurements, but the figure captions indicate that all yields presented in the manuscript were performed in triplicate or duplicate. All data were meant to be triplicate, but while filling out the required Reporting Summary, we realized that the the data in Figure 2A were only duplicate. This has been updated in the manuscript.

Reviewer #2 (Remarks to the Author):

Comment 1 (regarding optimization of reaction conditions; figure 1C)

From the text it is not clear if the improved reaction conditions with glutathione as HOBr scavenger were tested with other variants than the 4V+S. What about other mutants with promising yields and good e.e.s? Is this concept transferable?

The benefits of glutathione were observed for most enzyme/substrate pairs examined, but the reviewer is correct that we did not include those data. We added several representative examples in Fig. S14 and call out this figure in the last paragraph of the text before the summary. In all cases, enantioselectivity is comparable to or improved in the presence of glutathione. In some cases, like for 4V+S, the improvement is substantial.

Comment 2 (regarding chlorolactonization; line 96-98)

It would be nice to go into more detail, what that previous work (citation 23) showed about the halogen specificity. What could be the reason for the missing chlorolactonization activity on a molecular level? Do you think the main reasons is the distance of the carboxylate to the benzylic site?

As noted above, we have now found that chlorolactonization is possible using 4V+S, but only this enzyme and with significantly lower yield than the corresponding bromolactonization. While much of the FDH literature has focused on the control that different enzymes exert of halogen specificity (and that is certainly significant), I believe the inherent reactivity of different potential halogenating species doesn't get as much attention as it should. We have previously reported similar apparent specificity for aromatic bromination. To illustrate this point, we have repeated a couple specific examples from reference 25. I would prefer to leave this out of the SI since it is not directly related to halocyclization, but these results show that, in the presence of glutathione (thus no free HOX), a given halogenase can chlorinate or brominate some substrates but can only *brominate* others, and bromination typically occurs with higher yield than chlorination. Phenyl piperazine is a good example. We have never observed a FDH that can chlorinate or brominate some substrates but only *chlorinate* others. Observation of both chlorination or bromination means that both chloride and bromide are catalytically competent halide substrates, but the increased organic substrate specificity for chlorination suggests that the active halogenating agent is somehow less reactive for chlorination than for bromination (see more below).

Reference 36 indicates that HOBr forms more efficiently than HOCl prior to entering the active site of Thal, and a similar (35%) difference in decoupling for the two halide ions in 4V+S could explain why 4V+S chlorolactonization is less efficient than bromolactonization. As noted in ref 25, however, this difference in reactivity is also consistent with the known reactivity of nitrogen-substituted halogen reagents, such as haloimides, haloamines, etc., and can be rationalized in terms of the relevant electronegativity values: N, 3.04; O, 3.44; Br, 2.96; Cl, 3.16. These values indicate that an N-Cl bond is polarized as δ^+ , while the corresponding N-Br bond has the opposite polarization (δ^-), giving rise to a more electrophilic Br substituent and leading to apparent halogen specificity. The following statement was added to the manuscript to address a comment from reviewer 1, and I believe it addresses this comment also:

While preliminary evaluation of chlorolactonization of **1** using a subset of the enzymes ultimately screened for bromolactonization revealed no hits, subsequent analysis of 4V+S revealed that it catalyzes chlorolactonization of **1** to give **1c** in 30% yield and ~90:10 e.r. (Fig. S5). The lower chlorolactonization yield is consistent with our previous observation that aromatic bromination of non-native substrates is generally more prevalent and higher yielding than the corresponding chlorination,²⁵ and this promising activity bodes well for further directed evolution of FDHs with improved chlorolactonization activity.

Comment 3 (regarding unspecific reaction with HOX)

It is assumed that there is freely diffusible HOX released by the enzyme, which leads to an unspecific reaction outside of the active site and produces a racemic mixture. It would be preferable to write a few more sentences about free HOCl. Can it also be released? Why was no racemic mixture of the chlorolactonization product detected?

This is related to a question that reviewer 1 asked, and it is now addressed in Figure S9/10 and the accompanying text. In short, we do not see any background chlorolactonization for any of the K79A mutants studied, but we do see a decrease in enantioselectivity in the absence of glutathione. As I note in the text now included in the SI (below), it is not clear how to resolve these pieces of data. As I mentioned in response to reviewer 1, I think further detailed studies on relative rates for different processes catalyzed by different mutants will be required to resolve it, and I hope we can agree that lies beyond the scope of this initial study.

Chlorolactonization was only observed for 4V+S, and while similar yields were obtained under all conditions evaluated, a significant reduction in enantioselectivity was observed in the absence of glutathione. It is notable that no chlorolactonization is observed for 4V+S K79A (or RebH K79A) in the absence of glutathione at pH 7.6 or 6.8. On the other hand, 4V+S only provides 35% yield for chlorolactonization in the absence of glutathione. A similar reduction in yield for the corresponding K79A variant as observed for bromolactonization (80%) would lead to a chlorolactonization yield of around 7%. The expected [HOX]/[XO⁻] ratio is ~15 times higher for X=Br than for X=Cl at the pH values used, which could explain the lack of racemic background reaction observed for the latter. It is not currently clear why chlorolactonization enantioselectivity decreases in the absence of glutathione since there does not seem to be sufficient racemic background reaction based on the 4V+S K79A results to account for the observed decrease.

Reviewer #3 (Remarks to the Author):

1. Manuscript title: The title says “directed evolution” which implies that the paper includes an engineering study. Yet I cannot find any protein engineering performed in this study. If I am not mistaken, the enzyme variants reported here (apart from the control mutant K79A) come from a previously established panel, mainly a substrate walking approach previously published by the group (ref. 24) as well as genome mining (ref. 23). If engineering efforts contributing to improved halocyclization activity were performed herein, this must be mentioned more clearly in the manuscript (and also added to the experimental section). Otherwise I suggest considering a different title.

Reviewer 1 also raised this issue. The title was changed as noted above.

2. The authors use the term ‘yield’ when referring to biotransformation efficiency. Do they refer to ‘isolated yields’ or ‘conversions’ based on HPLC traces? Yield usually implies that a reaction product is isolated (at least on a milligram scale), purified and structurally characterised, e.g. by NMR. I suppose this has not been done here as the biotransformations were carried out on a scale of 70 μ L. Thus, ‘conversion’ seems to be more appropriate to me as the compounds have not been isolated. I recommend revising this throughout the manuscript.

I agree that precise definitions of these terms are important, and we did not do a good job of that in this manuscript. That being said, I would prefer to use the term “yield” since we are reporting product yield albeit via the use of HPLC assay rather than isolation. The following qualifier was added to the Figure 1 caption to clarify exactly what “yield” means: “**Product assay yields (hereafter, “yields”)** and selectivities are the average of triplicate measurements determined by LC/MS using *p*-bromoanisole internal standard.” Additional clarification of how assay yields were obtained was also added in the SI as noted below. I generally consider “conversion” to be a particularly nebulous term. It can refer to conversion of starting material and may be unadjusted for differential response of different species involved in a reaction, both of which would be inaccurate in the current case. One instance of the term “conversion” was removed from the manuscript.

3. One of the most interesting achievements is that reaction optimisation successfully improves the enantioselectivity of halocyclization by quenching HOBr through a change of pH, glutathione addition, or increased substrate loading. This is particularly astonishing since for FDHs (unlike haloperoxidases) the release of HOBr from the active site has not been observed before. The authors claim that this suggests a finding to be taken into account for future investigations on FDHs, e.g., when expanding the substrate scope (conclusion, lines 227-230). I understand that this is an important statement requiring further evidence, particularly because it is one of the key findings towards higher reaction selectivity in this study. Therefore, the authors could provide further data if the non-specific release of HOBr also impacts ‘native halogenation’, e.g., of indole(s), anilines, or if it is a side effect merely important for non-native catalysis. Moreover, the authors’ data suggest that the K79A mutant is able to release HOBr which is reasonable as the halogenating species is formed in the flavin-binding site and does not involve K79. I wonder if this mutant leads to detectable (trace) halogenation as well. These

interesting observations need further clarification prior publication because they closely relate to some basic investigations on halogenase catalysis by the groups of van Pee and Walsh.

At the reviewers request, we screened five electron rich aromatic substrates and found that two are brominated to a significant extent (15-20% conversion) by RebH K79A (see Fig. S13). Both substrates examined provide two brominated products but only one chlorinated product. The ratio of the brominated products is identical (~1:1) for RebH K79A and 4V+S K79A, but 4V+S gives different selectivity, consistent with the proposal that free HOBr, rather than the native halogenating species, is responsible for bromination in the case of the K79A mutants. No significant chlorination was observed for the K79A variants. These results suggest that HOBr leaked from FDH active sites can brominate substrates, but the extent to which this occurs will depend on the enzyme and the substrate used. This is now noted in the manuscript with a call to the SI:

Indeed, we confirmed that two electron rich aromatic substrates could be brominated using RebH K79A (Fig. S13), consistent recent work on aromatic halogenation by the FDH Thal³⁶. Release of HOBr thus appears to be exaggerated in the absence of K79, but this process can be mitigated by tuning the active site so that olefin substrates react before HOBr is released, as in 4V+S, or by reducing the reactivity of HOBr in solution using increased pH or added glutathione (Fig. S14).

4. The manuscript requires more details on how biotransformations were analysed. I assume RP-HPLC(-MS) was utilised to analyse the enzyme panels. How were the products confirmed? Did the authors compare HPLC traces with authentic standards? The Supporting Information contains synthetic details on substrate synthesis. Were the (racemic) lactone products synthesised as well?

All of the required data were provided in the SI, but clear direction on how it was used was not provided. The following text was added to each biocatalysis section of the SI to address this comment and the reviewer's concern about the definition of "yield" as noted above: Product (P) assay yields (AY) were determined relative to internal standard (IS) with appropriate response factors (RF) using the formula $AY = RF \cdot (\text{integral}_P / \text{integral}_{IS}) + C$. RF values are provided in section VII. Authentic racemic products were used for RF determination were prepared as described in section VI, and HPLC traces for the racemic products (needed to identify product enantiomers in bioconversions) and bioconversions are provided in section VIII.

5. It would be a good improvement if the authors could comment on absolute configurations in addition to enantiomeric ratios where possible, although I understand that this is hard to decipher. For clarity to the readers, please indicate chiral centres with an asterisk.

Asterisks were added. We were only able to find in the CCDC the crystal structure of one of the compounds (**3**) that we were able to synthesize, so I am only comfortable commenting on the absolute configuration for that structure. This was noted in the manuscript: "While low enantioselectivity for the *p*-fluorine-substituted substrate **3** was observed (Fig. 1D), the major product had the *S*-configuration based on comparison to authentic enantio-enriched product, providing experimental support for the simulated binding poses (Figure S6)." We added the additional note in the SI that "Reference 3 reported the crystal structure of the R-enantiomer of

compound **3**. HPLC analysis of the racemic compound in that report using the same column used in our analysis indicated that the R-enantiomer eluted before the S-enantiomer. Because we see that the first peak is the larger of the two, we conclude that 4PL+E423D provides predominately the R-enantiomer of compound **3**.”

6. Does background halogenation activity occur as a competing reaction, for instance, on the benzene moiety? As the halogenated products and the halolactone have got an identical mass further commenting is necessary. Also HPLC diagrams showing product formation could be added.

An early version of this manuscript (available on ChemRxiv) addressed this issue, but we removed the comment for brevity. A similar comment has now been added following discussion of substrate scope: “Aromatic halogenation has not observed for any of the substrates examined, consistent with the fact that FDHs typically only halogenate substrates with greater electronic activation than that imparted by one or two alkoxy substituents.³³” Representative HPLC chromatographs for each reaction were provided in Section VIII of the SI. The chromatograph for the racemic sample synthesized as described in section VI is provided (“Racemic”) followed by that for the bioconversion (“bioconversion”).

*7. To complete the story, it is important to run representative transformations on a low milligram scale. In this case, the purified products can be structurally characterised (NMR, HR-MS etc.) to finally confirm bromolactonization. I am aware that running larger-scale reactions is challenging using halogenases. However in light of the good conversions achieved for substrate **1** using mutant 4V+S, for instance, I am confident that this will be feasible on a scale of 5 mg. Moreover, this will further corroborate the applicability of halocyclization in synthesis of complex small molecules which will strengthen the impact of the manuscript.*

This should have been done, and we have now included these data for Substrate **1** using 4V+S as suggested. The product was isolated in 84% yield and 95:5 e.r., and details regarding the scale up procedure were added to the SI.

- Figure 1 contains too many details from different experiments. I suggest splitting the figure into 2-3 individual figures / tables and place them in the text where mentioned. For example: (A) // (B+D)// (C). Also the table in Figure (D) should be enlarged.

We agree with this sentiment. We went with (A) // (B+C) // (D). If the manuscript is accepted, we will work with the editors to make sure that (D) is the correct size.

- Figure 2B: Residue S111 is hardly visible.

I looked at a lot of different views of this structure, and any attempts to better see S111 ended up hiding residues or substrate features that are more important. No change was made.

- References: The authors should add more references from other groups who contributed to enzymatic halogenation in the past years.

In response to the reviewers' comments, a few additional references were added. The manuscript now contains 17 references by other groups on enzymatic halogenation, including at least one citation for each enzyme reported to catalyze halocyclization. The manuscript also contains 7 citations of studies from my own group, 6 of which are essential since they describe the original directed evolution or genome mining studies that led to the enzymes used in the current study (the other is a review that uniquely covers aspects of FDH mutagenesis). If the reviewer can provide specific examples of statements that are missing appropriate citations, we are happy to add those.

- line 89-98: *Please comment on the way of analysis. Adding an HPLC diagram, e.g., a time course (start + end point) might be a suitable illustration.*

This was addressed in response to the related comment above (clarification of the assay yield determination process). A specific call to the SI has now been added in the manuscript.

- line 109: *“pH from 7.4 to 9 (Fig. 1C, entries 1-4), albeit at the cost of product yield.” – Does saponification of the lactone occur at pH 9?*

We don't see any new products by LC/MS, so this does not appear to be occurring to a significant extent; we see more unreacted starting material at higher pH.

- line 136: *“Electrophilic attack of the olefin by a bromamine species (...)” – as the authors know it is ambiguous if the bromoamine actually acts as the halogenating species (Flecks et al. DOI: 10.1002/anie.200802466). Most likely HOBr performs the attack. Probably this is also the case for halocyclization.*

As the reviewer notes, the nature of the active halogenating species in FDH catalysis is currently ambiguous, so it would be best to avoid saying that either a bromamine or HOBr “likely” performs the attack. We updated the sentence to note attack by “**either a bromamine species or HOBr**” and included relevant citations that were included earlier in the manuscript.

-line 157: *“and variants 4V+S, 4PL E461G, and 7L possess in common active site mutations at only these residues” – E461G is also located in/ very close to the active site as it coordinates Trp-NH2 in the native reaction.*

This is true, but not all of the most active variants (4V+S, 4PL E461G, and 7L) possess this mutation. The idea behind this sentence was that since only those mutations are shared “they play particularly important roles in halocyclization catalysis”. Please let me know if further clarification is needed.

-line 140: *‘was obtained’ ; refers to ‘mixture’ (singular)*

- S5, line 177: *spelling ‘plate’ / ‘was’*

- S6, line 197: *spelling ‘reagent’*

- S6, line 202: *spelling ‘plate’ / ‘was’*

- S23, line 582: *please correct product of Wittig reaction to alkene*

- S23, line 590: *‘substituted’ (no capital letter)*

- S24, line 610: *Please correct the reaction product (alkene)*

All of the above were corrected as suggested. Several other typos were also corrected in the synthetic procedures.

Reviewers' Comments:

Reviewer #1:

Remarks to the Author:

Clarity of the manuscript has improved. The major contribution of this work is on the use of RebH mutants in combination with reaction optimization (use of glutathione and control of pH) to catalyze enantioselective olefin halocyclization.

However, some of the statement can mislead readers, I suggest the following modifications.

1. The demonstration of free HOBr release in FDH has been documented previously. The authors should refer to the previously published evidence properly.

1.1

-Page 7, "...Indeed, we confirmed that two electron rich aromatic substrates could be brominated using RebH K79A (Fig. S13), consistent recent work on aromatic halogenation by the FDH Thal...."

Suggestion

"...Indeed, we confirmed that two electron rich aromatic substrates could be brominated using RebH K79A (Fig. S13), consistent with the recent work which showed that free diffusible HOBr can be released from the active site of aromatic FDH Thal..."

1.2

-Conclusion "Uncovering this new manifold of FDH reactivity also led to the finding that FDHs can release freely diffusible HOBr from their active sites, indicating that caution is warranted when evaluating the activity of FDHs toward non..."

Suggestion

"..Uncovering this new manifold of FDH reactivity also led to the finding that RebHs and possibly other types of FDHs can release freely diffusible HOBr from their active sites, indicating that caution is warranted when evaluating the activity of FDHs toward non...."

2. Some of the statements referring to enzymatic halocyclization may be overstated. I would suggest toning it down.

2.1 Abstract

"Catalytic enantioselective halocyclization of alkenes is a powerful bond-forming tool in synthetic organic chemistry and a key step in natural product biosynthesis, but no examples of enantioselective enzymatic halocyclization of simple achiral olefins have been reported."

As VHPOs can also catalyze stereospecific halocyclization, I suggest modifying this sentence.

"Catalytic enantioselective halocyclization of alkenes is a powerful bond-forming tool in synthetic organic chemistry and a key step in natural product biosynthesis. Finding proper biocatalysts and optimizing reaction conditions capable of enantioselective enzymatic halocyclization would be useful for development of green chemistry in this area. Here, we ..."

2.2 Conclusion

"In summary, we have characterized the first examples of olefin halocyclization catalyzed by an FDH and the first examples of enantioselective halocyclization on simple, achiral substrates catalyzed by any enzyme..."

Consider removing "...catalyzed by any enzyme..."

Reviewer #2:

Remarks to the Author:

I looked carefully at the changes made by the authors and they addressed all questions raised by the reviewers in a solid and convincing way. I therefore fully support publication of these interesting results in Nature Communications.

Reviewer #3:

Remarks to the Author:

The manuscript entitled "Flavin-Dependent Halogenases Catalyze Enantioselective Olefin Halocyclization" reports on the first example of non-native biocatalysis using flavin-dependent halogenases. Starting from enzyme identification the authors conducted a stepwise optimisation and demonstrated its applicability on a small array of styrene derivatives, yielding halolactones in good to excellent enantioselectivities.

The previously mentioned issues were thoroughly addressed and resolved. The results are conclusive and well presented. The current manuscript is an excellent piece of work that considerably expands the toolbox of biocatalysis providing a starting point towards manifold applications in synthesis. Therefore I recommend publishing in Nature Communications.

Christian Schnepel

REVIEWERS' COMMENTS

Reviewer #1 (Remarks to the Author):

1.1

-Page 7, “..Indeed, we confirmed that two electron rich aromatic substrates could be brominated using RebH K79A (Fig. S13), consistent recent work on aromatic halogenation by the FDH Thal....”

Suggestion

“..Indeed, we confirmed that two electron rich aromatic substrates could be brominated using RebH K79A (Fig. S13), consistent with the recent work which showed that free diffusible HOBr can be released from the active site of aromatic FDH Thal...”

1.2

-Conclusion “Uncovering this new manifold of FDH reactivity also led to the finding that FDHs can release freely diffusible HOBr from their active sites, indicating that caution is warranted when evaluating the activity of FDHs toward non...”

Suggestion

“..Uncovering this new manifold of FDH reactivity also led to the finding that RebHs and possibly other types of FDHs can release freely diffusible HOBr from their active sites, indicating that caution is warranted when evaluating the activity of FDHs toward non....”

These changes were made with slightly different wording.

2.1 Abstract

“Catalytic enantioselective halocyclization of alkenes is a powerful bond-forming tool in synthetic organic chemistry and a key step in natural product biosynthesis, but no examples of enantioselective enzymatic halocyclization of simple achiral olefins have been reported.”

As VHPOs can also catalyze stereospecific halocyclization, I suggest modifying this sentence.

“Catalytic enantioselective halocyclization of alkenes is a powerful bond-forming tool in synthetic organic chemistry and a key step in natural product biosynthesis. Finding proper biocatalysts and optimizing reaction conditions capable of enantioselective enzymatic halocyclization would be useful for development of green chemistry in this area. Here, we ...”

2.2 Conclusion

“In summary, we have characterized the first examples of olefin halocyclization catalyzed by an FDH and the first examples of enantioselective halocyclization on simple, achiral substrates catalyzed by any enzyme...”

Consider removing “..catalyzed by any enzyme..”

The manuscript carefully describes current knowledge of stereoselective halocyclization:

“Some vanadium haloperoxidases (VHPOs) that catalyze stereoselective halocyclization have been characterized, but only native or putative substrates containing stereogenic centers have been reported to date.¹⁶⁻¹⁸ Other VHPOs^{19,20} and heme haloperoxidases (HHPOs)²¹ can catalyze halocyclization of synthetic substrates, but racemic products are obtained, likely due to their formation via free HOX generated by the enzyme. No examples of FDH-catalyzed halocyclization or enantioselective halocyclization involving simple, achiral substrates using any enzyme have been reported.”

Both of the statements noted by the reviewer were qualified to emphasize the unique reactivity of FDHs on simple achiral substrates relative to VHPOs, and a clear description of known VHPO reactivity with citations was provided. The distinction is important, but it would not be clear to those who only read the abstract. We have therefore taken the reviewer's suggestion regarding the abstract (the sentence before the one noted by the reviewer was also changed so that the suggested change fit with the rest of the abstract) but left the conclusion as originally written.